# Attention mechanism-based deep supervision network for abdominal multi-organ segmentation

Peng An[1][0009−0007−9335−3778] , Yurou Xu[1][0009−0005−4451−2774], and Panpan Wu[1∗][0000−0003−2915−2086]

Tianjin Normal University, China  {Panpan Wu}pwu@tjnu.edu.cn

**Abstract.** In this paper, we present a novel approach to multi-organ segmentation in abdominal CT examinations conducted across multiple centers, various phases, different vendors, and diverse disease conditions. This novel approach use deep learning(DL) and attention .We describe the strategy employed during the Fast and Low GPU Memory Abdominal Organ Segmentation (FLARE) challenge, which was held in conjunction with the International Conference on Medical Image Computing and Computer Assisted Intervention (MICCAI) 2023.To meet the challenge requirements and achieve faster model convergence within the specified time frame, we developed a U-Net architecture. Our U-Net is based on a lightweight network from the VGG family, serving as the encoder, with additional attention mechanisms incorporated into the decoder. The decoder is designed symmetrically to fully leverage forward skip connections. Attention modules were not only integrated within the decoder but also introduced before the final segmentation layer.With this strategy, it enables the model to converge in a short time and has a shorter number of iterations, in order to better cope with the time constraints of the competition. According to challenge rules, encoder and decoder weights are randomly initialized, without relying on any pretraining scheme. To improve the gradient flow and encourage extracting discriminative features, our model leverages multi-stage deep supervision for automatic depiction of tumors and 13 organs such as the liver, right kidney, spleen, etc., providing a new perspective for the interpretation and decision-making of clinical upper abdominal images.Our method achieved an average DSC score of 41.1% and 15.04% for the organs and lesions on the validation set and the average running time and area under GPU memory-time cure are 189s and 405109MB, respectively.

**Keywords:** Multi-organ Segmentation · DL · U-Net · VGG · Attention.

## 1 Introduction

The recent development of non-invasive imaging technologies has opened new horizons in studying abdominal structures. Segmentation has become a crucial task in abdominal image analysis with many applications such as computer-assisted diagnosis, surgery planning, imageguided intervention or radiotherapy

[30]. In particular, precise delineation of abdominal solid visceral organs, including the liver, kidneys, spleen, pancreas, and other organs, from Computed Tomography (CT) images, is of critical importance for localization, volume assessment, or follow-up purposes. However, the analysis of abdominal imaging datasets is challenging and time-consuming for clinicians, given the complexity of the abdominal region. Robust and automatic abdominal multi-organ segmentation is required to guide image interpretation, facilitate decision-making, and improve patient care, all while minimizing manual delineation efforts.

In this area, many interactive, semi- and fully-automated methods have been proposed with diverse methodologies including statistical shape models [2], multi-atlas segmentation [34] or machine learning [8] [7]. Outstanding performance has been reached in almost all medical image analysis tasks using deep learning (DL) [18]. Despite the large variability in organ shape, size, location and texture, abdominal multi-organ segmentation has naturally benefited from this massive trend [27] [12] [6]. Compared to conventional machine learning, the need for hand-crafted features no longer remains necessary. In particular, huge efforts have been devoted to automatic segmentation based on variants of Fully Convolutional Networks (FCN) [19]. In the medical image processing community, U-Net [26] is one of the most well-known approach among existing convolutional encoder-decoders. Able to learn from relatively small datasets, U-Net and its derivatives are the most likely to automatically infer high-level knowledge involved by radiologists when interpreting abdominal images.

The Fast and Low GPU Memory Abdominal Organ Segmentation (FLARE) challenge, organized in conjunction with the International Conference on Medical Image Computing and Computer Assisted Intervention (MICCAI) 2023, is an extension of the official competitions held in 2021 and 2022. Due to time constraints, our team is particularly interested in the model [4] that achieved eighth place in the 2021 competition. It is impressive that it managed to secure the eighth position despite its modest configuration, small model size, and low GPU usage. Therefore, our team aims to leverage the attention mechanism, which enhances local features, to improve the convergence speed of their model without significantly affecting its accuracy. In light of our late entry into the competition, our goal is to develop a multi-organ deep learning architecture with faster convergence speed and acceptable accuracy.

## 2   Method

To ensure faster convergence of the model while maintaining its original segmentation efficiency, we build a U-Net architecture based on a lightweight VGG-13 network from the VGG family [28] as encoder. The decoder adds attention [24] and is constructed in a similar way to obtain a symmetrical construction while keeping long-range shortcuts [5] [6] and before the last layer of the neural network, attention is added to obtain a faster Rate of convergence. According to the challenge rules which prevent relying on any pre-training scheme, weights of both encoder and decoder branches are randomly initialized. To further enhance

performance, our model, as illustrated in Figure 1, leverages multi-stage deep supervision [33] [9] and incorporates attention mechanisms [24].

## 2.1 Preprocessing

Intensity normalization is used as pre-processing step.Thus, each CT volume is clipped to the [1, 99] percentiles of the intensity values. In addition, a z-score normalization is applied based on the mean and standard deviation of the intensity values among the whole training dataset. Neither cropping nor resampling is employed. Not using Data clean also not conducting statistical analysis.

## 2.2 Proposed Method

Network architecture: Our model comprises an encoder-decoder architecture with forward skip connections from the encoder stages to their corresponding decoder stages. In contrast to the standard U-Net [26], we utilize a simpler yet effective VGG-13 encoder with batch normalization layers (torchvision.models.vgg13_bn).

To avoid large GPU memory consumption, we designed a 2D multi-class segmentation model with C = 15 classes dealing with background (bg), liver (li), right kidneys (r-ki), spleen (sp) , pancreas (pa), aorta (at), inferior vena cava(i-v-c), right adrenal gland (r-a-g), left adrenal gland (l-a-g), gallbladder (gb), esophagus (ep), stomach (sm), duodenum(dd), left kidney (l-ki), tumor (tm). The network independently processes axial slices to produce 2D segmentation masks which are then stacked together to recover 3D volumes. To exploit spatial relationships between abdominal structures, the model learns to simultaneoulsy delineate the multiple organs instead of relying on several organ-specific models.

The basic layer pattern consists of sequential layers including $3\times3$ convolutional layers (conv) with $1\times1$ stride and $1 \times 1$ padding followed by batch normalization (BN) and Rectified Linear Unit (ReLU) activation. Such pattern is repeated twice and followed by $2 \times 2$ max pooling (MP). The encoder comprises a sequence of 4 [conv, BN, ReLU]x2 + MP patterns (Fig. 2). The first convolutional layer generates 64 channels. The number of channels doubles after each MP layer until it reaches 512. Compared to VGG-13 [28], top layers including fully-connected layers and softmax are omitted. The fifth [conv, BN, ReLU]x2 pattern from original VGG-13 serves as central part to separate contracting and expanding paths.

To get a symmetrical construction while still using forward skip connections, the decoder branch is extended in the similar fashion as the encoder by adding batch normalization layers and more features channels [6] and adding attention mechanism(ReLU+Sigmoid) after concatenate operation (Fig. 2).Additionally, feature maps as outputs of each intermediate decoder blocs are upsampled using bilinear interpolation to the size of the input image. In the same spirit as in [33] [9], a convolutional operation with $3 \times 3$ kernel is applied to create 16

feature maps at each level (Fig. 2). These maps then go through deep supervision modules to improve the gradient flow and encourage learning more useful representations [9]. After having performed the concatenation of intermediate outputs (Fig. 1), two convolutional layers including a final one with using attention and softmax activation achieves pixel-wise multi-label segmentation.

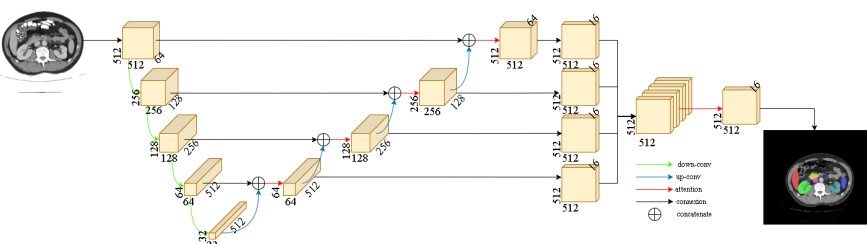

**Fig. 1.** Overall model framework.In this framework, some repetitive and non-critical details are omitted. Each convolutional block consists of multiple convolution layers. Max-pooling (MP) is used for downsampling, and transposed convolution is used for upsampling. Before upsampling, an attention calculation is performed on the concatenated feature maps, and then upsampling is carried out. Feature matrices obtained from different depth supervision are directly concatenated, followed by attention calculation and convolution to reduce the number of channels.

**Loss function.** Our network is trained with the cross-entropy loss function $L_{ce}$ defined below:

$$L_{ce} = \frac{1}{N} \sum_{c=1}^{C} \sum_{i=1}^{N} g_i^c log p_i^c$$

where $N$ is the number of pixels in the axial slices. $p_i^c$ and $g_i^c$ denote respectively the predicted probability and ground truth at pixel $i$ for class label $c \in \{$ bg, li, r-ki, sp, pa, at, i-v-c, r-a-g, l-a-g, gb, ep, sm, dd, l-ki, tm $\}$.

The overall loss function $L$ is the average sum of the cross-entroy losses estimated at different decoder levels involving supervision:

$$L = \frac{1}{M+1} \sum_{j=1}^{M} L_{ce}^j + L_{ce}^f$$

where $L_{ce}^j$ denote the loss for the points of supervision at $j$ layer of the decoder. Following the VGG-13 architecture [28], $M = 4$ intermediate decoder levels are considered. $L_{ce}^f$ correspond to the loss computed at the final network output ( $f$ stands for final). Note that level $j = 1$ is closer to the network ending part than level $j > 1$. Because the original model and code used multiple GPUs for deep supervision processing, but our team only had a single GPU to run the program, we made changes to the overall loss function. And after improvement, the performance is similar or even better than the original model, so this method is useful when a single GPU is needed to run. We did not apply the Dice loss function during backpropagation, but rather used it as a visualized indicator.

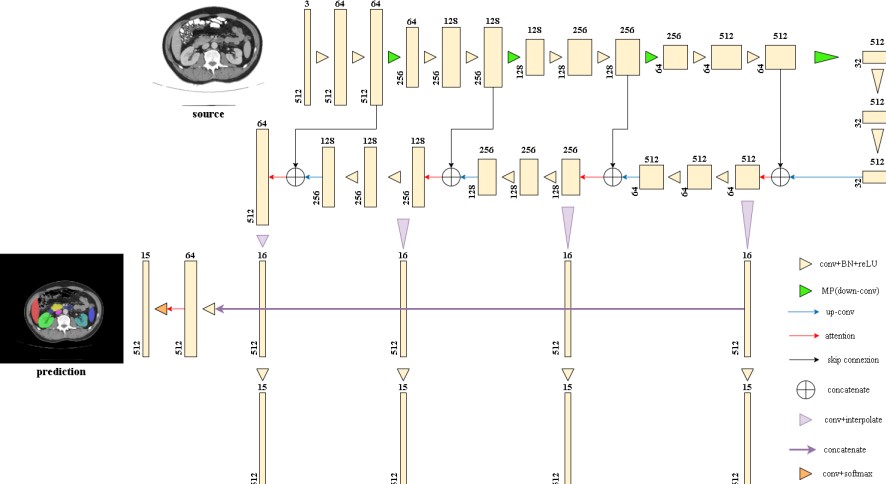

**Fig. 2.** Detailed convolutional encoder-decoder architecture. In this context, 'up-conv' refers to transposed convolution. 'Attention' is a combination of the ReLU and Sigmoid functions. 'Interpolate' is performed using the bilinear interpolation function.

With this indicator and the cross-entropy loss, we performed parameter tuning and other tasks.

**Number of model parameters.** Number of model parameters. The number of trainable parameters is 24,496,331 (around 93.5Mb), much less than the 41,268,192 parameters employed in nnU-Net [16].

**Unlabeled images were not used and not used the pseudo labels generated by the FLARE21 winning algorithm [15] and the best-accuracy-algorithm [31].**

### 2.3   Post-processing

As post-processing, we keep the largest connected segmented areas for voxels respectively labeled as 13 organs, for example, liver, spleen and pancreas etc. No ensembling method is used.

## 3   Experiments

### 3.1   Dataset and evaluation measures

The FLARE 2023 challenge is an extension of the FLARE 2021-2022 [21][22], aiming to aim to promote the development of foundation models in abdominal disease analysis. The segmentation targets cover 13 organs and various abdominal lesions. The training dataset is curated from more than 30 medical centers under the license permission, including TCIA [3], LiTS [1], MSD [29], KiTS [13,14],

autoPET [11,10], TotalSegmentator [32], and AbdomenCT-1K [23]. The training set includes 4000 abdomen CT scans where 2200 CT scans with partial labels and 1800 CT scans without labels. The validation and testing sets include 100 and 400 CT scans, respectively, which cover various abdominal cancer types, such as liver cancer, kidney cancer, pancreas cancer, colon cancer, gastric cancer, and so on. The organ annotation process used ITK-SNAP [35], nnU-Net [17], and MedSAM [20].

The evaluation metrics encompass two accuracy measures—Dice Similarity Coefficient (DSC) and Normalized Surface Dice (NSD)—alongside two efficiency measures—running time and area under the GPU memory-time curve. These metrics collectively contribute to the ranking computation. Furthermore, the running time and GPU memory consumption are considered within tolerances of 15 seconds and 4 GB, respectively.

### 3.2   Implementation details

**Environment settings** The development environments and requirements are presented in Table 1.

**Table 1.** Development environments and requirements.

| System | Ubuntu 18.04.6 LTS |
|---|---|
| CPU | Intel(R) Xeon(R) Silver 4210 CPU @ 2.20GHz |
| RAM | 16×4GB |
| GPU (number and type) | 0 NVIDIA GeForce RTX 2080 Ti |
| CUDA version | 11.3 |
| Programming language | Python 3.7 |
| Deep learning framework | torch 1.7.0+cu92, torchvision 0.8.1+cu92 |
| Specific dependencies | scikit-image, nibabel, torch |
| Code | https://github.com/0NGU0/FLARE2023-PengAn.git |

**Training protocols** Our team utilized only 2200 partially labeled CT scans during the training phase. In the validation stage, we followed a method where 10% of the cases were randomly selected from the pool of 2200 CT scans with partial labels for model validation. The test set was not separated from the initial 2200 labeled CT scans.

Data augmentation methods were not employed in this model, but dropout was utilized. In selecting the optimal model, we used the results submitted to the competition website as the benchmark for determining the optimal weight generation and model selection. This benchmark was primarily based on the DSC (Dice Similarity Coefficient) and NSD (Normalized Surface Dice) indicators for tumor assessment in the website results.

**Table 2.** Training protocols.

| | |
|---|---|
| Network initialization | normal initialization |
| Batch size | 2 |
| Patch size | 512×512 (full axial slices) |
| Total epochs | 5 |
| Optimizer | Adam |
| Initial learning rate (lr) | $10^{-5}$ |
| Lr decay schedule | no decay |
| Training time | 27 hours per iteration and the 5 generation took a total of 135 hours |
| Loss function | |
| Number of model parameters | 93.55M |
| Number of flops | 212.95G |
| $CO_2$eq | / |

## 4   Results and discussion

From the current situation, the improved model has achieved similar or better results with fewer iterations and less time.

Possible reasons for segmentation errors or incompleteness include:

The model may not have fully converged, and certain features may not have been learned effectively. Although improvements have been made in the convergence speed of the model, the original model's limitations in segmenting closely located organs have not been addressed. Due to the limited number of tumor cases in the training set and the random occurrence of actual tumors, the model's performance in predicting tumors may not achieve the same level of segmentation accuracy as for other organs.

### 4.1   Quantitative results on validation set

In Table  3, we present the Dice and NSD scores for the validation of specific organs and tumors. We did not utilize unlabeled data for training, validation, or prediction. This decision was made because when our team became aware of the competition, a month had already passed since it began. To expedite all stages of the process, we opted to use only 2200 CT scans with labeled data for training. (This choice was made because more raw data would require additional preprocessing time, and we needed to experiment with various preprocessing strategies.) As a result, no ablation experiments were conducted to investigate this aspect.

### 4.2   Qualitative results on validation set

Fig.3 depicts two simple cases (upper part) and two challenging cases (bottom part) from the validation set. The source axial slices, ground truth, and predicted

**Table 3.** Quantitative evaluation results.

| Target | Public Validation | | Online Validation | | Testing | |
|---|---|---|---|---|---|---|
| | DSC(%) | NSD(%) | DSC(%) | NSD(%) | DSC(%) | NSD (%) |
| Liver | 93.27 ± 2.78 | 92.01 ± 2.96 | 87.91 | 86.94 | 61.39 | 58.68 |
| Right Kidney | 86.8 ± 2.42 | 86.18 ± 2.54 | 86.37 | 85.27 | 62.14 | 62.17 |
| Spleen | 89.54 ± 4.36 | 89.41 ± 4.64 | 84.64 | 84.41 | 60.92 | 60.1 |
| Pancreas | 71.39 ± 1.6 | 83.79 ± 1.61 | 69.69 | 81.86 | 48.14 | 55.63 |
| Aorta | 15.31 ± 7.57 | 15.62 ± 7.64 | 16.86 | 17.46 | 7.96 | 7.29 |
| Inferior vena cava | 12.54 ± 6.63 | 12.26 ± 6.25 | 18.31 | 17.62 | 7.27 | 6.49 |
| Right adrenal gland | 4.22 ± 3.96 | 5.05 ± 5.43 | 12.87 | 16.85 | 4.9 | 5.78 |
| Left adrenal gland | 2.11 ± 2.26 | 2.48 ± 3.04 | 7.14 | 9.26 | 3.28 | 3.51 |
| Gallbladder | 17.95 ± 7.88 | 17.65 ± 7.68 | 29.1 | 28.72 | 17.47 | 16.68 |
| Esophagus | 10.76 ± 4.92 | 14 ± 6.42 | 13.05 | 16.47 | 5.24 | 5.76 |
| Stomach | 10.2 ± 4.09 | 10.77 ± 4.32 | 10.85 | 11.53 | 3.3 | 3.14 |
| Duodenum | 6.6 ± 4.75 | 10.92 ± 4.96 | 13.27 | 17.49 | 4.94 | 6.09 |
| Left kidney | 83.38 ± 3.75 | 83.14 ± 3.72 | 84.18 | 84.76 | 62.81 | 63.47 |
| Tumor | 12.56 ± 2.69 | 7.69 ± 1.58 | 15.04 | 9.23 | 8.21 | 4.17 |
| Average | 36.9 ± 4.26 | 37.93 ± 4.49 | 41.1 | 42.97 | 25.57 | 25.64 |

**Table 4.** Quantitative evaluation of segmentation efficiency in terms of the running them and GPU memory consumption. Total GPU denotes the area under GPU Memory-Time curve.

| Case ID | Image Size | Running Time (s) | Max GPU (MB) | Total GPU (MB) |
|---|---|---|---|---|
| 0001 | (512, 512, 55) | 39.59 | 2218 | 75269 |
| 0051 | (512, 512, 100) | 60.07 | 2218 | 121299 |
| 0017 | (512, 512, 150) | 98.89 | 2218 | 206949 |
| 0019 | (512, 512, 215) | 122.14 | 2218 | 258027 |
| 0099 | (512, 512, 334) | 185.31 | 2218 | 397158 |
| 0063 | (512, 512, 448) | 307.15 | 2218 | 665363 |
| 0048 | (512, 512, 499) | 347.57 | 2218 | 754546 |
| 0029 | (512, 512, 554) | 351.28 | 2218 | 762258 |

label maps are presented from left to right. The liver, kidneys, spleen, pancreas, and other organs are color-coded in red, green, blue, yellow, and other colors, respectively. Taking into consideration the limited computational resources, our model demonstrates strong performance in segmenting larger or distinct organs. However, in cases where diseased tissues are present, or when certain organs are in close proximity to others, the segmentation effectiveness correspondingly diminishes.

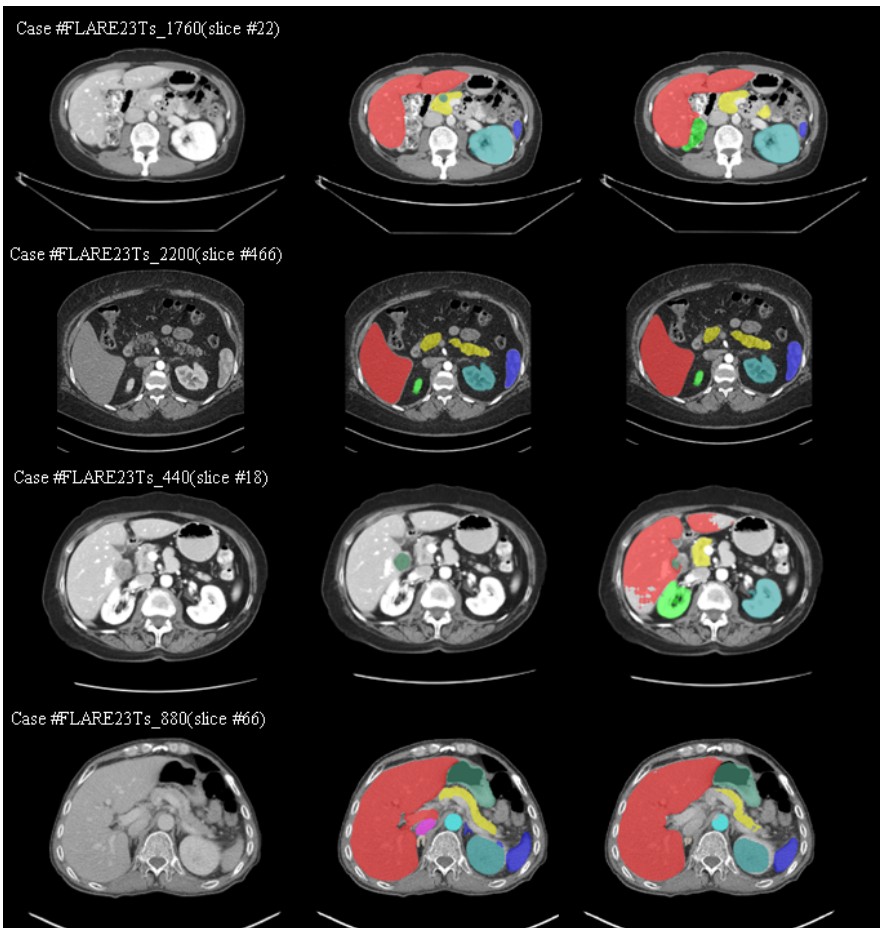

**Fig. 3.** Source axial slices, ground truth and predicted label maps are shown from left to right.

**Table 5.** The table shows the data comparison between our improved model and the unmodified model.The data in the table comes from the submission results of the website.

| Model Category | Organ | | Tumor | | Iterations |
|---|---|---|---|---|---|
| | DSC(%) | NSD(%) | DSC(%) | NSD(%) | |
| Origin | 32.95 | 33.31 | 6.03 | 1.6 | 1 |
| Origin | 39.23 | 40.44 | 8.13 | 2.76 | 3 |
| Origin | 43.9 | 45.35 | 8.1 | 3.29 | 5 |
| Origin | 38.24 | 39.29 | 11.83 | 7.03 | 7 |
| Origin | 40.64 | 42.07 | 15.23 | 9.58 | 9 |
| Add attention | 40.43 | 41.54 | 14.12 | 8.42 | 1 |
| Add attention | 35.97 | 37 | 11.61 | 7.03 | 2 |
| Add attention | 40.16 | 41.58 | 7.69 | 4.9 | 3 |
| Add attention | 37.18 | 38.39 | 14.32 | 8.86 | 4 |
| Add attention | 41.1 | 42.97 | 15.04 | 9.23 | 5 |

### 4.3 Segmentation efficiency results on validation set

Based on the data presented in Table 5, our model performs equally well or better than the eighth-place model within the same total time frame.

### 4.4 Results on final testing set

The final test results, including DSC and NSD metrics, are shown in the 'Testing' column of Table 3. The average time for case predictions is 83.03 seconds, and the GPU memory usage is 169,244. Although the final test results are lower compared to the online validation results, this was expected. On the one hand, the uncertainty in the location and shape of tumors has led to a decrease in tumor prediction accuracy. On the other hand, the lack of model convergence and poor generalization have resulted in reduced accuracy in the segmentation of non-tumor organs.

### 4.5 Limitation and future work

This model did not fully converge, and the prediction time for larger cases exceeded 60 seconds due to personal and time constraints. However, the results from the previous iterations demonstrate the effectiveness of the improvements. When compared to the previous model, it achieves better or similar results with fewer iterations and shorter processing times. In the future, as the model achieves full convergence, the obtained results can be reevaluated, and additional horizontal comparative experiments can be conducted to fully demonstrate the method's effectiveness and generalizability. Furthermore, optimization can be applied to the models used for prediction, ensuring that each case can be processed within 60 seconds, regardless of its size.

## 5   Conclusion

The multi-center, multi-phase, multi-vendor, and multi-disease CT data were segmented using deep learning and an attention mechanism. This work demonstrates that there is still room for improvement in the convergence speed of The eighth model of flare competition in 2021, under the same number of iterations, and our team has provided a solution for this. Standard pipelines have been extended to lightweight convolutional encoder-decoders with deep supervision and attention mechanisms. Preliminary results suggest that the attention we have incorporated has played a significant role in accelerating the model's convergence, thereby avoiding the need for resource-intensive computational processes in clinical practice. While our approach accurately processes many images containing healthy organs or organs with small lesions, the presence of large tumoral areas is a critical factor affecting delineation performance. Additionally, the segmentation task for the pancreas and other organs warrants further investigation to enhance the capacity of deep learning models in handling the substantial inter-patient anatomical variability in terms of size, shape, location, and texture.

**Acknowledgements** The authors of this paper declare that the segmentation method they implemented for participation in the FLARE 2023 challenge has not used any pre-trained models nor additional datasets other than those provided by the organizers. The proposed solution is fully automatic without any manual intervention. We thank all the data owners for making the CT scans publicly available and CodaLab [25] for hosting the challenge platform.

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

**Table 6.** Checklist Table.

| Requirements | Answer |
| --- | --- |
| A meaningful title | Yes |
| The number of authors ($\leq 6$) | 3 |
| Author affiliations, Email, and ORCID | Yes |
| Corresponding author is marked | Yes |
| Validation scores are presented in the abstract | Yes |
| Introduction includes at least three parts: background, related work, and motivation | Yes |
| A pipeline/network figure is provided | Figure 1&2 |
| Pre-processing | Page 3 |
| Strategies to use the partial label | Page 6 |
| Strategies to use the unlabeled images. | Page 7 |
| Strategies to improve model inference | Page 3 |
| Post-processing | Page 5 |
| Dataset and evaluation metric section is presented | Page 5 |
| Environment setting table is provided | Table 1 |
| Training protocol table is provided | Table 2 |
| Ablation study | Page 7 |
| Efficiency evaluation results are provided | Table 3 |
| Visualized segmentation example is provided | Figure 3 |
| Limitation and future work are presented | Yes |
| Reference format is consistent. | Yes |