# OpenReview forum: "Attention mechanism-based deep supervision network for abdominal multi-organ segmentation"
_MICCAI.org/2023/FLARE — Submitted to FLARE 2023_

### Official Review · Reviewer_6WM4 · 2023-10-04
**MICCAI FLARE23 challenge-Attention mechanism-based deep supervision network for abdominal multi-organ segmentation**

**Rating:** 6
**Confidence:** 5

**Review:**

This paper designed an approach based on the VGG family networks, performing slice-wise predictions and achieving mediocre scores on the quantitative evaluation results. In addition, the paper performs at parr with running times requirements. A key list of things to consider: 1. The VGG network did not use pre-trained weights, was not pre-trained in a self-supervised manner, 2. Only uses Cross-Entropy loss (and not CE and Dice Loss), 3. Does not use unlabeled images nor any pseudo labels provided by FLARE 23 organizers (winners of previous competitions).

The paper also states that the model did not fully converge, and hence it will be exciting to read on how the model flares as the above mentioned points are used together.

---

> ### Author Response · Authors · 2023-11-14
>
> We sincerely thank you for your encouraging and constructive comments, which has boosted our confidence in our work. Regarding the points of interest you raised, we would like to provide more detailed explanations.
>
> Comment 1: The VGG network did not use pre-trained weights, was not pre-trained in a self-supervised manner.
>
> Response: Because our main focus is on how to make the model converge faster within the same timeframe, we did not perform any work related to pre-training.
>
> Comment 2: Only uses Cross-Entropy loss (and not CE and Dice Loss).
>
> Response: Although we only applied the cross-entropy loss function to the calculation of backpropagation, we used the Dice loss function as a visualized indicator to observe whether the model was trained appropriately. We performed parameter tuning and other tasks using this indicator and the cross-entropy loss, which can be found in our code. Additionally, we have added supplementary descriptions in Section 2.2 of our paper.
>
> Comment 3: Does not use unlabeled images nor any pseudo labels provided by FLARE 23 organizers (winners of previous competitions).
>
> Response: Due to our late participation in the competition, we didn't have time to process the provided pseudo labels and unlabeled images during the competition. Now we are working on this, but we haven't made much progress, so we haven't reflected it in the paper.

---

> > ### Comment · Reviewer_6WM4 · 2023-11-27
> > **Reply**
> >
> > Hello,
> >
> > That is perfectly fine - I do hope you achieve more gains by using pretraining and pseudo labels. All the best!

---

### Official Review · Reviewer_KY2S · 2023-10-04
**MICCAI FLARE23 challenge-Attention mechanism-based deep supervision network for abdominal multi-organ segmentation**

**Rating:** 5
**Confidence:** 4

**Review:**

In this paper, they applied the attention technique to the decoder of a U-Net-based model and introduced a strategy to construct a VGG-13 network from the VGG family. It is interesting to note that they experimented with using the attention technique to improve accuracy and speed.

---

> ### Author Response · Authors · 2023-11-14
>
> We thank you for your comments on our paper, which have provided us with a better understanding of the paper's content.
>
> Although our results were not as ideal because within the limited time for the competition, we didn’t have sufficient time to wait for the the model converging, we are still confident in our improvements. As the data presented in Table 5 indicates: 'This modification allows the network to better suppress irrelevant regions in the input image while highlighting useful features in the decoder stage, and it enables the network to achieve the same results in a shorter time.' In the following time, we will continue to delve deeper into the current work, such as model convergence and areas that need refinement, and we believe we can achieve even better and more exciting results in future works.
>
> Thank you once again for your feedback.

---

### Official Review · Reviewer_vDWN · 2023-10-25

**Rating:** 8
**Confidence:** 5

**Review:**

Don't need to include "MICCAI FLARE23 challenge" in the tile since this work will be published in the miccai flare23 proceeding.

Fig. 1-2. Please provide an overview description of the network.


# Review 4

1. The font in the picture should be Times New Roman.
2. The figure resolution is low.

---

> ### Author Response · Authors · 2023-11-14
>
> We sincerely thank you for your valuable and constructive comments. After reflecting on these suggestions, we have revised the manuscript. The modifications made can be summarised as follows:
>
> Comment 1: Don't need to include "MICCAI FLARE23 challenge" in the title since this work will be published in the miccai flare23 proceeding.
>
> Response: The title has been changed from "MICCAI FLARE23 challenge-Attention mechanism-based deep supervision network for abdominal multi-organ segmentation" to "Attention mechanism-based deep supervision network for abdominal multi-organ segmentation."
>
> Comment 2: Fig. 1-2. Please provide an overview description of the network.
>
> Response:  Overviews for Fig. 1 and Fig. 2 have been provided.
>
> Comment 3: The font in the picture should be Times New Roman.
>
> Response:  The font in the figures has been changed to Times New Roman.
>
> Comment 4: The figure resolution is low.
>
> Response:  The resolution of the images has also been increased.
>
> Once again, we appreciate your feedback.

---

### Decision · Program_Chairs · 2023-10-25

Accept